# A New Species *Amecephala micra* sp. nov. (Hemiptera: Liadopsyllidae) from Mid-Cretaceous Myanmar Amber [note 1]

**DOI:** 10.3390/insects16030302

**Published:** 2025-03-13

**Authors:** Jowita Drohojowska, Marina Hakim, Diying Huang, Jacek Szwedo

**Affiliations:** 1Institute of Biology, Biotechnology and Environmental Protection, University of Silesia, 9, Bankowa Street, 40-007 Katowice, Poland; jowita.drohojowska@us.edu.pl; 2State Key Laboratory of Palaeobiology and Stratigraphy, Nanjing Institute of Geology and Palaeontology, Chinese Academy of Sciences, Nanjing 210008, China; marina@nigpas.ac.cn; 3Laboratory of Evolutionary Entomology and Museum of Amber Inclusions, Department of Invertebrate Zoology and Parasitology, University of Gdańsk, 59, Wita Stwosza Street, 80-308 Gdańsk, Poland

**Keywords:** Sternorrhyncha, Liadopsyllidae, new species, Cretaceous, Myanmar amber

## Abstract

We have discovered and described a new species of extinct family Lia-dopsyllidae *Amecephala micra* sp. nov. It is a male insect that has been preserved as an inclusion in amber from Myanmar, originating from the mid-Cretaceous period. This insect has features that have never before been seen in Liadopsyllidae. These features include details of the antennae, hind legs, and hindwings. It has a row of short spines on the tip of the hind tibia, which suggests it could jump, and the strengthening of the posterior edge of the hind wing may be part of the equipment used for vibrational communication. These issues are discussed in relation to modern Psylloidea and Liadopsyllidae. This finding is an important addition to our knowledge of the differences and diversity of Liadopsyllidae and how they have evolved.

## 1. Introduction

The Sternorrhyncha, a suborder of the Hemiptera, comprises a number of highly prevalent and plant-damaging pests, including aphids, whiteflies, scale insects, and psyllids. Sternorrhyncha derive nutrition by extracting sap from plants, a process that can result in the gradual weakening and eventual demise of their hosts. In addition to this, they are responsible for transmitting plant viruses and facilitating the proliferation of fungal species such as sooty mould. The Sternorrhyncha, a name derived from Latin ‘sternum’ = chest, and Ancient Greek ‘rhynchos’ = snout, are distinguished by their piercing/sucking mouthparts, which originate from the posterior region of the head. These insects are characterised by their small size, soft body, and delicate nature, with membranous fore and hind wings. Notably, a significant proportion of these insects have evolved symbiotic relationships with bacteria, with numerous species also engaging in symbiotic interactions with ants [1,2,3]. The Sternorrhyncha, which comprise approximately 18,700 described extant species, have a long evolutionary history, with records reaching the Permian, and with a number of extinct lineages and groups [3]. The extinct sternorrhynchan family Liadopsyllidae is an opophagous hemipteran group related to modern psyllids. The fossil record of this family extends back to the Jurassic and Cretaceous strata, as well as to fossilized resins from the Toarcian–Turonian period. Since the formal establishment of the family, the taxonomic concept and content of the group have undergone significant variation, and a consensus regarding its classification remains elusive. The most recent phylogenetic analysis of the sternorrhynchans, including extinct groups, places them as a monophyletic unit in a sister position to modern psylloids, known since the Eocene [3,4]. Currently, nine genera and 23 species are attributed to this family, but some taxa require revisionary studies. The genus *Amecephala* Drohojowska, Szwedo, Müller et Burckhardt, 2020 was described from the mid-Cretaceous (Aptian-Cenomanian) amber of Kachin, Myanmar with a single species, *A. pusilla* Drohojowska, Szwedo, Müller et Burckhardt, 2020. The present study reports the second species of the genus from the aforementioned locality and age.

## 2. Materials and Methods

### 2.1. Geological Setting

The specimen being subject of this report is an inclusion in mid-Cretaceous, fossilised resin from the Kachin area in Northern Myanmar. In recent decades, Cretaceous fossil resin from Myanmar has become the most important source of organism inclusions, shedding new light on the Cretaceous Terrestrial Revolution through numerous discoveries of plants, invertebrates, and vertebrates [5,6,7,8,9,10,11]. Three main sources of amber have been reported in Myanmar. The first source was discovered in Hukawng Valley, Kachin State. It has been mined for centuries [12,13,14,15]. Amber mines are scattered throughout the area, including the Khanjamaw site, which is the largest colonial mine, the Noije Bum site—‘Banyan Mountain’ in the Jingpaw language [16], the Inzutzut site [17], the Angbamo site [18,19], and the Xipiugong site [20]. This resin has been mineralogically classified as burmite [21,22,23]. The amber-bearing deposits are clastic sedimentary layers with thin limestone beds and abundant coaly and carbonaceous material. The resin has been dated variously, but according to investigations by [24], it is 98.79 ± 0.62 Ma, which currently dates the deposit to the earliest Cenomanian. A slightly older age has been postulated based on the study of fossil ammonites and pholadid bivalve inclusions [16,25,26,27,28,29,30]; see also discussion of age in [31,32]. The discovery and subsequent exploitation of insectiferous amber from the Pat-tar Bum area near Khamti (Hkamti), in the Sagaing Region, has revealed a new era of research [33,34,35,36]. These appear to represent different assemblages. The amber is deposited in the tuffaceous rocks, with an age estimated radiometrically at 109.7 ± 0.4 Ma [37] or slightly older (~112 Ma; see [32]). The specific sites noted by [34] include Lachun (also spelled Laychun and Lachon) Maw, which was the most productive mining area, as well as Kyat Maw, Shan Maw, Gyar (also spelled Kyar) Maw, and Kyauk Tan Maw (‘maw’ means ‘mine’ in Burmese language). The amber from Hti Lin (Tilin) in the Magway Region of western central Myanmar [35,38] is also insectiferous, but distinctly younger, aged uppermost Campanian, ~72.1 Ma [29] based on tuff collected just above the amber-bearing layers. The presence of different sites of amber mines results in the commercial sources of ‘Myanmber amber’ being variable, and it is difficult to ascertain the exact age of Myanmar amber given that the ages of the deposits range from ~110 to ~72 Ma, especially for Hkamti and Kachin amber, which the FTIR-ATR spectra show little difference [34]. The differentiation of these fossil resins poses a significant challenge; consequently, the employment of advanced analytical instruments and methodologies (e.g., photoluminescence spectroscopy, solid ^13^C nuclear magnetic resonance spectroscopy, and Raman spectroscopy) has been imperative to achieve more precise characterisation of the resins. This assertion is supported by several previous works [39,40,41,42,43,44].

To avoid any confusion and misunderstanding, all authors declare that the sample reported in this study was legally collected before June 2017, and it was not involved in the armed conflict and ethnic strife in Myanmar [45]. The fossil specimen is deposited permanently in the Nanjing Institute of Geology and Palaeontology, Chinese Academy of Sciences, Nanjing, China (see ‘NIGP Statement & Museum Catalogue Entry’ in Appendix A), in full compliance with the International Code of Zoological Nomenclature [46], Statement of the International Palaeoentomological Society [47], and policies presented by Haug et al. [48]. The paper is registered in ZooBank under LSID:urn:lsid:zoobank.org:pub:B882476F-9D43-4AF7-9AE1-E1C64CEFCE9C.

### 2.2. Morphology and Documentation

Observations were performed using a Zeiss AXIO Zoom V16 stereomicroscope and a Zeiss AXIO Imager Z2 compound microscope (Carl Zeiss AG, Oberkochen, Germany), both equipped with digital cameras. Photomicrographs with green fluorescence were obtained with the compound microscope, which is connected to a laser source (eGFP mode; excitation/emission: 450–490/515–565 nm). Other photomicrographs were taken with a Zeiss LSM710 confocal laser scanning microscope (Carl Zeiss AG, Oberkochen, Germany), at a 488 nm Argon laser excitation line [49]. All photos were stacked using the Helicon Focus 6 software (Helicon Soft, Kharkiv, Ukraine). The illustrations were processed and arranged using the Adobe Photoshop CC 2019 software package (Adobe Inc., San Jose, CA, USA).

Morphological terminology follows mostly [50,51,52], but the interpretation of venation is treated in concordance with [53,54].

## 3. Results

### Systematic Palaeontology

Order Hemiptera Linnaeus, 1758 [55]

Suborder Sternorrhyncha Amyot et Audinet-Serville, 1843 [56]

Infraorder Psyllodea Flor, 1861 [57] [=Psyllaeformia Verhoeff, 1893] [58]

Family Liadopsyllidae Martynov, 1926 [59]


* *


**Genus †*Amecephala*** Drohojowska, Szwedo, Müller et Burckhardt, 2020 [4]

Type species. *Amecephala pusilla* Drohojowska, Szwedo, Müller et Burckhardt, 2020; by original designation.

Age and occurrence. late Albian–Cenomanian, late Lower/early Upper Cretaceous (‘mid-Cretaceous’ auct.); Kachin State, Myanmar.

Emended diagnosis. Vertex rectangular; coronal suture developed in the apical half; median ocellus on the ventral side of the head, situated at the apex of the frons, which is large, triangular; genae not produced into processes. Eyes hemispheric, relatively small, narrower than half of the vertex width (Figure 2A). Antenna with pedicel about as long as flagellar segment 1 (antennomere 1), pedicel longer than the remaining antennomeres (Figure 2B). Pronotum ribbon-shaped, relatively long, about ¾ as long as the vertex in the midline, laterally of equal length as medially. Forewing (Figure 3A) elongate, widest in the middle, pterostigma broad, subtriangular, not delimited at base by a vein; thus, vein R_1_ not developed; branches of vein M subequal in length; cell cu_1_ low and very long. Female terminalia short, cuneate. Male terminalia with distinct subgenital plate (hypandrium), parameres sickle-shaped and enlarged proctiger (Figure 5C,D).


* *



**†*Amecephala micra* sp. nov.**


urn:lsid:zoobank.org:act:26CA6E23-34DE-4E09-A7F0-DBE533B93AFF

(Figure 1, Figure 2, Figure 3, Figure 4, Figure 5 and Figure 6)

*Etymology*. From Ancient Greek adjective μικρός (mikrós), meaning very small, little, minute.

*Holotype*. Male, specimen number NIGP206856; deposited in the Nanjing Institute of Geology and Palaeontology, Chinese Academy of Sciences, Nanjing, China. Complete and well-preserved (Figure 1A–D) inclusion in burmite (Kachin amber).

*Locality and stratum*. Myanmar, Kachin State, Hukawng Valley, SW of Maingkhwan, former Noije Bum 2001 Summit Site amber mine (closed). Lowermost Cenomanian, Upper Cretaceous.

*Diagnosis*. Similar in appearance to *Amecephala pusilla* Drohojowska, Szwedo, Müller et Burckhardt, 2020, but differs in body size (body length 0.83 mm *A. micra*, body length 1.2 mm *A. pusilla*); *Amecephala micra* sp. nov. is the smallest Liadopsyllidae species known so far; anterior margin of the forewing almost straight (anterior margin of the forewing basally more arcuate in *A. pusilla*); terminal Rs slightly arcuate, ends at the margin apicad of half of the terminal M_1+2_ level (terminal Rs bent at the base and in the terminal section, reaching the margin basad of half of the terminal M_1+2_ length in *A. pusilla*); stem M relatively long, longer than forkings M_1+2_ and M_3+4_ (stem M shorter than forks M_1+2_ and M_3+4_ in *A. pusilla*), cell m shorter than cell m in *A. pusilla*; apex of the forewing lies in cell m_1_ near the apex of terminal M_1+2_ (in *A. pusilla*, the apex of the forewing shifted to terminal M_3+4_); terminal Cu_1b_ slightly recurrent, directed basad (terminal Cu_1b_ perpendicular to the tornus in *A. pusilla*); mesopraescutum wider than the long, subhexagonal, posterior margin straight in the median section (mesopraescutum subtriangular, almost twice wider than long in the middle, posterior margin arcuate in *A. pusilla*).

*Description*. Male; female unknown. Body minute, 0.83 mm long.

Head weakly inclined from the longitudinal body axis, about as wide as the pronotum and mesoscutum, head with compound eyes 0.28 mm wide, 0.1 mm long in the midline; vertex subrectangular, width at the posterior margin 0.14 mm; disc with the imbricate microsculpture visible; anterior margin arcuate, anterolateral angles widely angulate, lateral margins subparallel, posterior margin slightly concave; coronal suture developed in the apical half, basal half not visible; lateral ocelli on low eminences, placed closer to the anterior angles of the compound eyes; median ocellus on the ventral side of the head, situated at the margin of the frons and vertex, on a low eminence; frons reduced; antennal foveae (toruli) oval, slightly elevated; lower portions of face (genae) enlarged, triangular, not produced into processes; compound eyes hemispheric, relatively small, narrower than half of the vertex width (Figure 2A); antenna with 10 antennomeres (scapus, pedicel, and 8 flagellomeres), filiform, moderately long, scapus globular, pedicel cylindrical, elongate, about as long as the 1st flagellomere; flagellum 0.30 mm long; flagellum 1.07 times as long as the head width; rhinaria visible on the 2nd and 4th flagellomeres (Figure 2B); flagellomeres more slender than the pedicel, the 1st flagellomere the longest, the 2nd, 4th, and 6th flagellomeres and the 3rd, 5th, and 7th flagellomeres of similar lengths, relative lengths of flagellomeres counted from base: 1.0:0.6:0.5:0.6:0.65:0.6:0.5:0.8; clypeus elongate, about 4 times as long as wide in widest section, not swollen, postclypeus slightly wider in the upper section, with a rounded upper margin, anteclypeus slightly narrower (Figure 2C); loral plates distinct, crescent-shaped, upper angles slightly below the upper margin of postclypeus, lower angles not reaching the lower margin of anteclypeus; clypellus not visible, rostrum short, reaching the mesocoxae; apical segment shorter than the penultimate one.

**Figure 1 insects-16-00302-f001:**
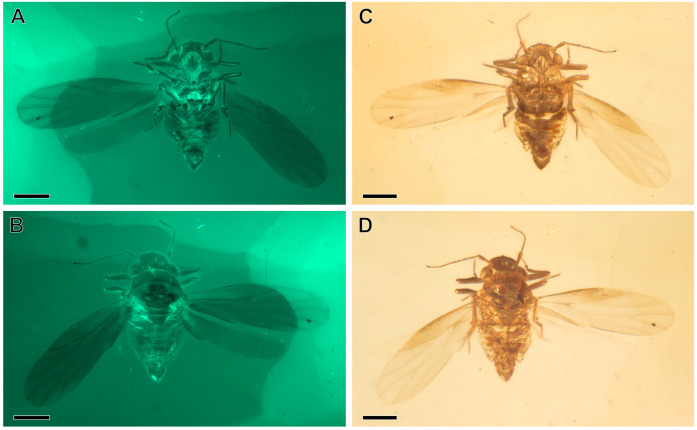
*Amecephala micra* sp. nov., holotype, NIGP206856: (**A**,**C**) Body in dorsal view. (**B**,**D**) Body in ventral view; Scale bars = 0.2 mm.

Thorax with the pronotum wider than the mesopraescutum and as wide as the mesoscutum, laterally of the same length as medially (Figure 2D); disc of the pronotum with imbricate sculpture. Mesothorax large; mesopraescutum subhexagonal, nearly 3 times wider than long in the midline, with a more delicate imbricate sculpture, anterior margin slightly arcuate, posterior margin with posterolateral edges oblique and converging, median section of the posterior margin straight; mesoscutum about as wide as the pronotum, shorter in the midline than the pronotum and the mesopraescutum, widely U-shaped; disc with imbricate sculpture anteromedian angles straight with apex rounded; anterolateral margin about as long as the mesopraescutum in the midline, the antero- and posterolateral margins forming an acute angle, posterolateral margin merely arcuate, posterolateral angles widely rounded, median section of the posterior margin shallowly concave; mesoscutellum narrow, band-like, anterior margin weakly concave, 0.6 times as long in the midline as the length of the mesopraescutum, lateral margins widely arcuately converging, posterior margin merely concave; metascutum narrower than the mesoscutellum, about as long in the midline as the mesoscutellum, metascutellum trapezoid, with disc imbricate, anterior margin straight, lateral margins converging the posteriad, posterior margin slightly arcuate.

Forewing (Figure 3A) 0.83 mm long, 0.26 mm wide, 3.2 times as long as wide; membrane transparent, colourless, veins pale; forewing elongate, narrow at the base, widest in the middle, narrowly rounded at the apex which lies in cell m_1_; costal margin straight, thickened, flattened, anteroapical angle widely rounded, apex round at the level of the ending of the terminal M_1+2_, posteroapical angle widely rounded, tornus almost straight, long, longer than claval margin, claval margin almost straight; slight incision (anal break) at claval apex visible; costal break not visible, most probably absent; pterostigma broad, subtriangular, not delimited at the base by a vein, thus vein R_1_ not developed; stem R slightly longer than the stem M + Cu; bifurcation of the vein R proximal to the middle of the wing; vein R_2_ relatively short, weakly sigmoidal, oblique, reaching the margin apicad of half of the forewing length, distinctly shorter than the branch Rs; branch Rs long, slightly curved towards the anterior margin, reaching the apicad of the terminal Cu_1a_; stem M long and straight, longer than stems R and M + Cu; stem M longer than its branches, terminals M_1+2_ and M_3+4_ of subequal length; stem Cu short, splitting into a very long Cu_1a_ and a short Cu_1b_, terminal Cu_1a_ slightly arcuate, reaching the margin at the posteroapical angle; cell r_1_ triangular; cell m_1_ value 2.6, cell cu_1_ (areola postica) low and very long; cell cu_1_ value more than 7.0; surface spinules not visible; claval suture visible (Figure 3A and Figure 4A); anal break near the apex of vein Cu_1b_.

**Figure 2 insects-16-00302-f002:**
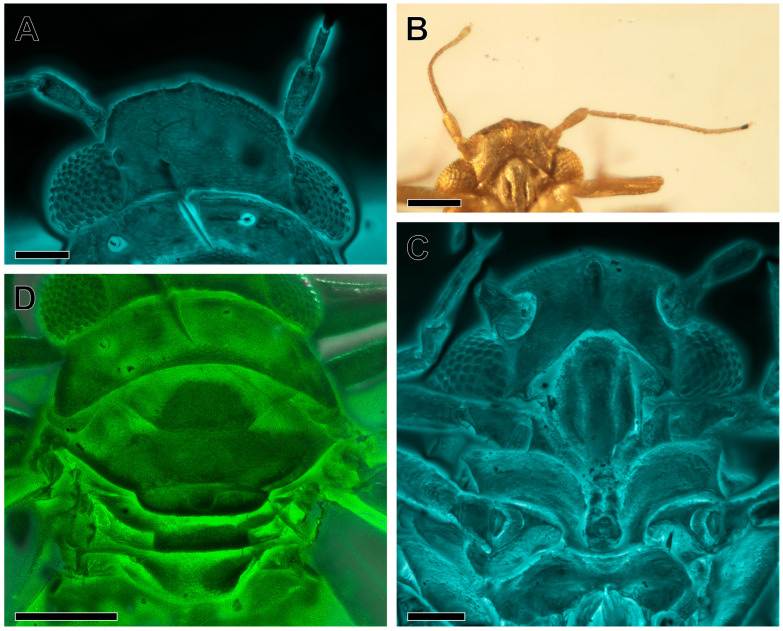
*Amecephala micra* sp. nov., holotype, NIGP206856: (**A**) Head in dorsal view, in confocal imagery. (**B**) Antennae in ventral view. (**C**) Head and mouthparts in ventral view, in confocal imagery. (**D**) Thorax in ventral view, in confocal imagery. Scale bars: 0.05 mm for (**A**,**C**); 0.1 mm for (**B**,**D**).

Hindwing (Figure 3B) shorter than the forewing, four times as long as wide, membranous, transparent, and colourless, veins pale; basal section of the costal margin with a few spines; costal margin bent at the base, then straight; anteroapical angle rounded, apex shifted anteriad, at the ending of Rs, posteroapical angle widely arcuate, deep claval incision/claval notch separating an elongate anal lobe with a strengthened posterior margin (stridulatory area?); stem R + M + Cu short, stem Cu first separated, at the level of the costal margin bent, stem R + M longer than the common stem R + M + Cu, longer than branch R; forking of branch R basad of half of the hindwing length; terminal R1 oblique, reaching margin at about 2/3 of the hindwing length, less than half of Rs length; terminal Rs slightly arcuate, reaching the margin near the apex of the hindwing (Figure 3B and Figure 4B); terminal M long, almost straight; terminal Cu sigmoid, long, reaching the margin apicad of half of the distance between the claval notch, claval fold weaker than the veins but distinct.

**Figure 3 insects-16-00302-f003:**
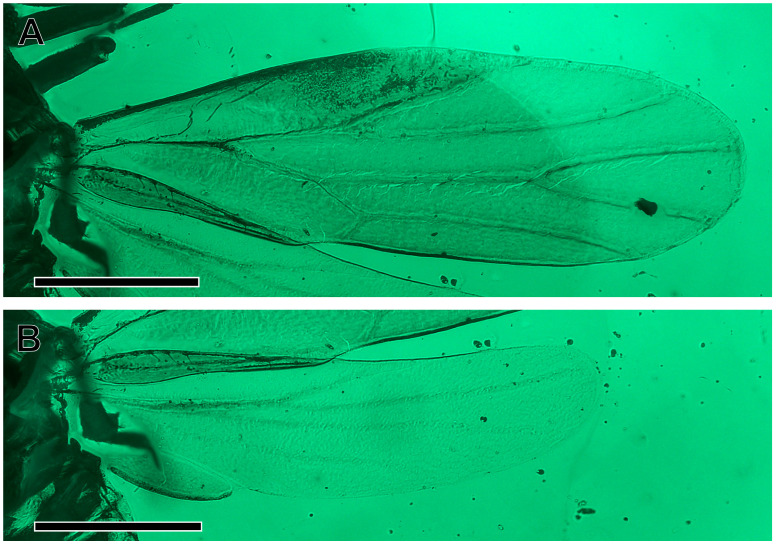
*Amecephala micra* sp. nov., holotype, NIGP206856: (**A**) Forewing, in green fluorescence imagery. (**B**) Hindwing, in green fluorescence imagery. Scale bars = 200 micron.

**Figure 4 insects-16-00302-f004:**
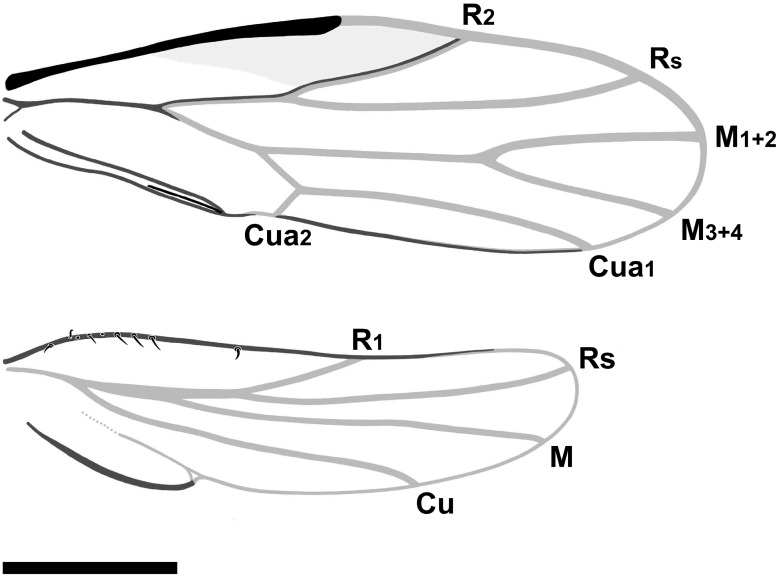
*Amecephala micra* sp. nov., holotype, NIGP206856: Line drawings of wings. Scale bar = 200 micron.

Legs similar in shape and size, moderately elongate (Figure 5A); profemora slightly thickened, protibia shorter than the profemora, protarsus about ¾ of the protibia length, basitarsomere subtubular, about half of the length of the distitarsus, tarsal claws long and slender, pulvilli absent; mesofemur similar in size and thickness to profemur, mesotibia about as long as the protibia, mesotarsus similar in structure to the protarsus; metacoxal subconical, without a mercanthus; metatrochanter ring-like, metafemur slightly thickened, but not distinctly stronger than the pro- and mesofemur, without a genual spine, widened apically with a row of 13–14 apical, short, sclerotized spurs (Figure 5A,B); metatarsi two-segmented, basitarsomere subtubular, elongate, with a subapical, oblique callosity on the plantar surface (?), about as long as the distitarsomere; distitarsomere without additional setae; tarsal claws long and slender, pulvilli absent (Figure 5B).

Abdomen widely connected to the thorax, the first two segments narrow, with the sternites fused (?), tergite 3 slightly enlarged, and tergites 4 to 8 of similar length (Figure 5C); sternites 3 to 8 constricted in median sections (Figure 5D). Proctiger longer than the subgenital plate, 10th abdominal segment separated, small, and placed terminally; subgenital plate subquadrate, with the posterior margin slightly concave; parameres sickle-shaped, with tips not exceeding the apical margin of the proctiger; connective subrectangular, widely transversely placed; aedeagus, thin, with a hinge separating it into basal and apical portions (Figure 5A,B and Figure 6).

**Figure 5 insects-16-00302-f005:**
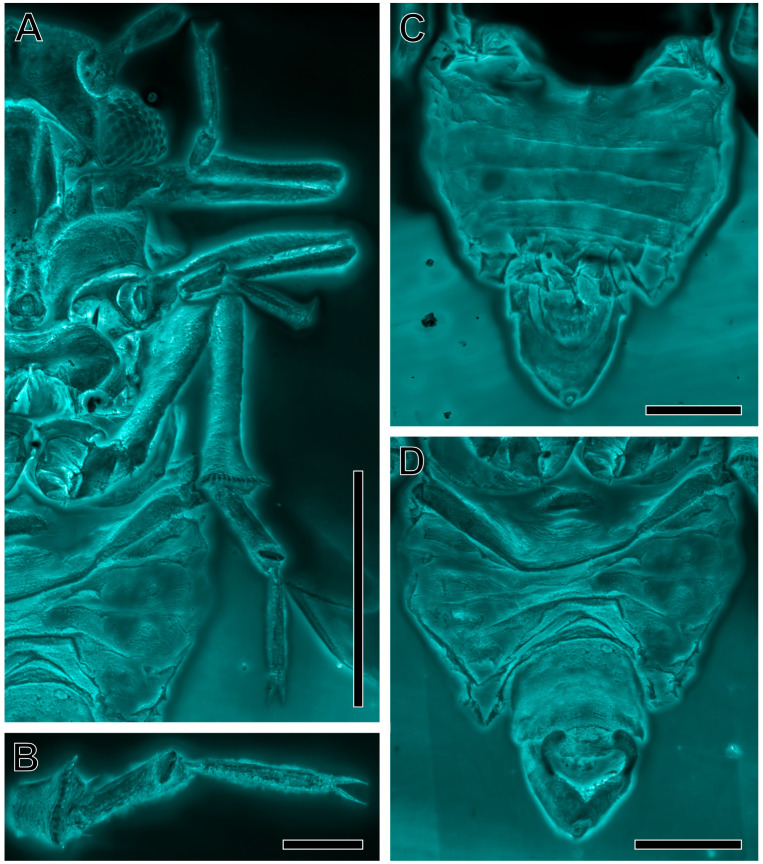
*Amecephala micra* sp. nov., holotype, NIGP206856: (**A**) Legs in ventral view, in confocal imagery. (**B**) Hind tarsus, in confocal imagery. (**C**) Abdomen and terminalia in dorsal view, in confocal imagery. (**D**) Abdomen and terminalia in dorsal view, in confocal imagery. Scale bars: 0.05 mm for (**B**); 0.1 mm for (**C**,**D**); 0.2 mm for (**A**).

**Figure 6 insects-16-00302-f006:**
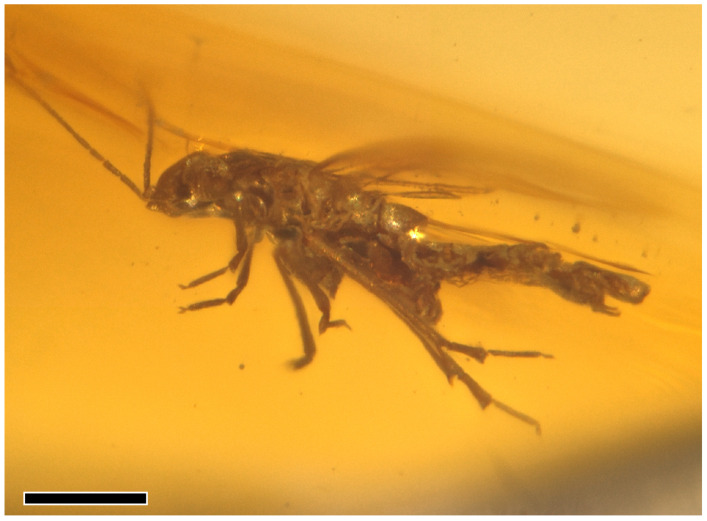
*Amecephala micra* sp. nov., holotype, NIGP206856: Body in lateral view; Scale bar = 200 micron.

## 4. Discussion

In the present study, we describe a new species of the extinct Liadopsyllidae genus *Amecephala* and provide diagnostic features and illustrations for the new taxon represented by the male specimen. The genus *Amecephala* differs from related Liadopsyllidae by the venation, with the forewing with pterostigma (lacking in *Liadopsylla*), veins R and M + Cu of the forewing subequal in length and vein Cu_1a_ distinctly curved (vein R of the forewing distinctly longer than M + Cu and vein Cu_1a_ almost straight in *Malmopsylla*), and cell cu_1_ low and very long, around 6.0 times as long as high (cell cu1 higher and shorter, less than 2.5 times as long as high in *Neopsylloides*, *Gracillinervia*, and *Pauropsylloides*). The terminalia of Liadopsyllidae are weakly recognised as these fossils are known in the vast majority from sedimentary rocks. The status of the genera *Mirala* and *Burmala* described from Kachin amber and previously placed in Liadopsyllidae [54,60] is controversial and is the subject of current revisions [61].

Special attention should be given to the hindwing of *Amecaphala micra* sp. nov. as this structure is very weakly known in the family Liadopsyllidae [53,62]. In *Amecephala pusila* (female), the hindwing is membranous, transparent, and colourless, with venation weakly visible, with vein Rs not reaching the margin, the claval fold partly visible, and a weak not emarginated claval margin [4]. In *Amecephala micra* sp. nov. (male), the venation is visible, with veins R, RS, M, and Cu, the claval fold distinct, and a clearly emarginated claval margin. These differences could be related to taphonomic reasons; however, it could also be the expression of sexual dimorphism of these insects. Adult males and females of modern Psylloidea are morphologically similar, except that males are often smaller, have different genitalia, and thus different abdominal shapes. The differences may be expressed by small morphological differences in coloration, the legs’ armature, morphometrics of the wings, or olfactory structures of the antennae [63,64,65,66,67,68,69], and it can be assumed that similar differences had to be present among related, extinct Liadopsyllidae. Stridulation by psyllids was first reported by Ossiannilsson [70] and Heslop-Harrison [71], and recently overviewed by Taylor [72], Liao et al. [73,74], and Avosani et al. [73,74,75]. Psyllids also use their host plants as a medium for sexual communication with vibrational signals transmitted through the substrate, as well as wing buzzing [76]. Stridulitrum on the mesothorax–plectrum on the axillary cord was postulated as a mechanism [73,75,77,78], but this opinion was challenged by Poljanar et al. [76]. It is questionable whether the strengthening of the claval margin of the hindwing of *Amecephala micra* sp. may be related to the stridulatory behavior as in modern psyllids, however this hypothesis appears to be plausible. The lobe of the hindwing and the strengthening of the claval margin similar to the hindwing of *Amecephala micra* sp. nov. is observed, e.g., in *Trichochermes* Kirkaldy, 1904 [79,80] or *Togepsylla* [81], but the function of this structure remains enigmatic.

*Amecephala micra* sp. nov. is classified within the Liadopsyllidae, among other features, on the absence of the meracanthus on the metacoxa [60,82]. Presence of the meracanthus on the metacoxa is a feature associated with jumping abilities, with a set of characters such as the enlargement and thickening of the meron, the backward twisting of the plane of the trochanter, femur, and tibia, and the enlargement of the trochanteral tendon [52,81]. This complex set of features is, however, variously present among various taxa of modern psyllids, with varying jumping abilities [81]. Jumping abilities are also expressed by the relative size of the pro- and mesofemora, as well as the thickening of the metatibia and the presence of a genual spine and an apical row of spurs on its tip [83]. Interestingly, in *Amecephala micra* sp. nov., the pro- and mesolegs are of similar length and structure, with the basitarsomere shorter than the apical tarsomere, and the metalegs are not distinctly different in size and thickness. However, the apex of the metatibia is provided with a row of short and thick apical setae (not typical spurs as in Psylloidea), and the tip of the basimetatarsomere is also thickened, with the basitarsomere as long as the apical tarsomere. Such a set of features may suggest that *Amecephala micra* sp. nov. could be similar in jumping abilities to the half-jumping lice *Togepsylla* Kuwayama, 1931 (Aphalaridae) [84], as interpreted by Luo et al. [81]. This will be an interesting step in furthering our understanding of the evolution of jumping abilities and their associated mechanisms in Psylloidea.

The newly described species *Amecephala micra* sp. nov. is an interesting and important fossil, but the knowledge of Liadopsyllidae and relative groups preserved as inclusions in fossil resins as well as depressions in rock matrix preclude us at the moment from further nomenclatorial decisions.

## 5. Conclusions

The findings and detailed descriptions of fossils are of extreme value for taxonomic and palaeobiodiversity studies. The incorporation of inclusions in fossil resins has the potential to contribute novel data for taxonomic classification, while concurrently facilitating an enhanced comprehension of morphological disparity and morphofunctional analyses. The fossils described above offer a novel perspective on evolutionary traits, palaeodiversity, disparity, and the relationships of the ancient psyllomorphous family Liadopsyllidae and their modern counterparts.

## Data Availability

The original contributions presented in this study are included in the article. Further inquiries can be directed to the corresponding author.

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
