# Peer review of "A New Species Amecephala micra sp. nov. (Hemiptera: Liadopsyllidae) from Mid-Cretaceous Myanmar Amberâ€"

_insects, 2025, doi:10.3390/insects16030302_

Round 1
Reviewer 1 Report
Comments and Suggestions for Authors
No comments. My recomendation is to publish it
Author Response
Comments 1: No comments. My recomendation is to publish it
Response 1: Thanks a lot.
Reviewer 2 Report
Comments and Suggestions for Authors
This paper describes a new hemipteran species, Amicephala micra (family Liadopsyllidae), from Cretaceous Myanmar amber. Its miniature body size, hind tibial apical spines, and specialized wing venation provide novel insights into the evolution and paleoecological adaptations of Sternorrhyncha insects. Suggestions:
- There some mixed use of ‘Myanmar amber’ , ‘Burmese amber’, 'burmite' in the title and main text, please using ‘Myanmar amber’ or ‘ Kachin amber’ rather than ‘Burmese amber’ .
- In the methods, 'To avoid any confusion and misunderstanding, all authors declare that the sample 102 reported in this study was legally collected before June 2017’, you can cite paper doi.org/10.1038/s41559-021-01479-z to strengthen it.
Author Response
Comments 2:
- There some mixed use of ‘Myanmar amber’ , ‘Burmese amber’, 'burmite' in the title and main text, please using ‘Myanmar amber’ or ‘ Kachin amber’ rather than ‘Burmese amber’ .
- In the methods, 'To avoid any confusion and misunderstanding, all authors declare that the sample 102 reported in this study was legally collected before June 2017’, you can cite paper doi.org/10.1038/s41559-021-01479-z to strengthen it.
Response 2: It is checked and the relevant terms changed. The suggested paper add, thanks.
Reviewer 3 Report
Comments and Suggestions for Authors
This is a simple article describing a new species. It contributes to the fossil record. The article is written in good English and is understandable to the reader. The manuscript is well illustrated.
My complaints concern the Discussion. Here, comparison, justification of systematic placement and the discussion itself are mixed.
Authors should provide Comparison and Systematic placement after describing a new species. It is necessary to indicate on the basis of what features it was placed in the family and genus.
Author Response
Comments 3: This is a simple article describing a new species. It contributes to the fossil record. The article is written in good English and is understandable to the reader. The manuscript is well illustrated.
My complaints concern the Discussion. Here, comparison, justification of systematic placement and the discussion itself are mixed.
Authors should provide Comparison and Systematic placement after describing a new species. It is necessary to indicate on the basis of what features it was placed in the family and genus.
Response 3: Yes it is, thanks for the comments.; We disagree, as we follow the rules and logic sequence of descriptive papers: emended diagnosis for the genus is given and we do not intend to change or discuss its family placement, as it was done in paper describing the genus, in our opinion repetition here is not necessary; Differential diagnosis for the new species is given with distinguishing characters presented and compared; in our opinion it is not necessary to make repetitive narration of distinguishing features again, just to make the discussion longer.